# MotiMotion: Motion-Controlled Video Generation with Visual Reasoning

**Lee Hsin-Ying** [1] [*]   **Hanwen Jiang** [2]   **Yiqun Mei** [2]   **Jing Shi** [2]   **Ming-Hsuan Yang** [1]   **Zhixin Shu** [2]

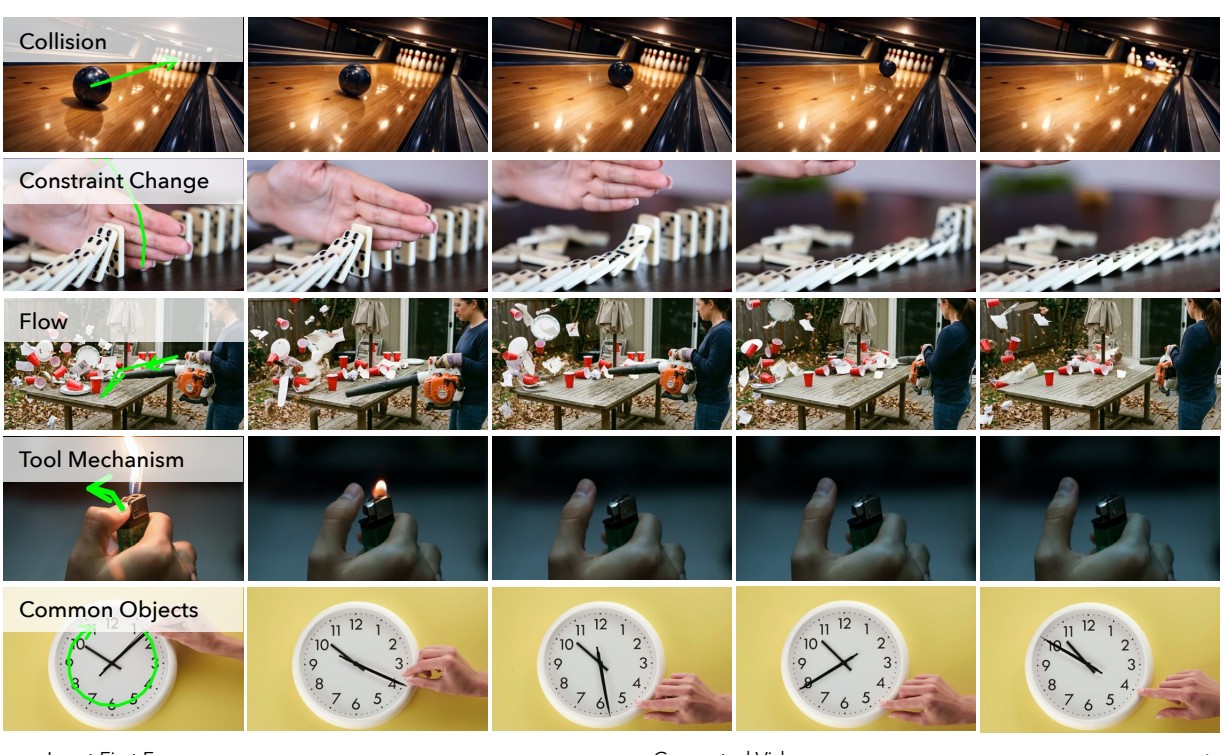

*Figure 1.* **MotiMotion.** We devise a motion-controlled video generation model that enables intelligent and natural interaction. Given sparse, raw trajectories (visualized in green lines) and prompts from users, MotiMotion reasons about the intention of inputs and predicts subsequent events that align with world knowledge, common sense, and physics principles. We demonstrate MotiMotion's capability to process diverse scenarios, including *collision*, *constraint change*, *flow*, *tool mechanism*, and *common objects*. The framework understands visual context and produces realistic content. Project page: https://motimotion.github.io

## Abstract

Current motion-controlled image-to-video generation models rigidly follow user-provided trajectories that are often sparse, imprecise, and causally incomplete. Such reliance often yields unnatural or implausible outcomes, especially by missing secondary causal consequences. To address this, we introduce MotiMotion, a novel framework that reformulates motion control as a reasoning-then-generation problem. To encourage causally grounded and commonsense-consistent interactions, we leverage a training-free vision-language reasoner to refine image-space coordinates of primary trajectories and to hallucinate plausible secondary motions. To further improve motion naturalness, we propose a confidence-aware control scheme that modulates guidance strength, enabling the model to closely follow high-confidence plans while correcting artifacts under low-confidence inputs with its internal generative priors. To support systematic evaluation, we curate a new image-to-video benchmark, MotiBench, consisting of interaction-

[*]Work done as an intern at Adobe Research [1]University of California, Merced [2]Adobe Research. Correspondence to: Zhixin Shu <zshu@adobe.com>.

*Proceedings of the 43$^{rd}$ International Conference on Machine Learning*, Seoul, South Korea. PMLR 306, 2026. Copyright 2026 by the author(s).

centric scenes where new events are triggered by motion. Both VLM-based evaluation and a human study on MotiBench demonstrate that MotiMotion produces videos with more plausible object behaviors and interaction, and is preferred over existing approaches.

## 1. Introduction

The field of image-to-video generation has been driven by the rapid maturation of diffusion models (Ho et al., 2020; Sohl-Dickstein et al., 2015) and the emergence of large-scale foundation models capable of synthesizing high-fidelity, temporal dynamics (DeepMind, 2025c; OpenAI, 2024; Wan et al., 2025). While these models demonstrate unprecedented visual quality and semantic alignment with textual descriptions, they face a critical bottleneck in practical applications: precise, logical controllability. Unlike static image generation, where spatial arrangement is often sufficient, video generation requires control over temporal evolution, necessitating an understanding of world knowledge, such as physics, causality, and object properties.

To address this, motion control enables users to provide explicit guidance, such as drag trajectories (Wu et al., 2024c; Yin et al., 2023), bounding box sequences (Wang et al., 2025b; 2024a), or flow maps (Burgert et al., 2025; Zhang et al., 2025c), to direct the generation process. These approaches bridge the gap between static prompts and dynamic output by allowing users to "draw" the motion. However, this paradigm relies on an assumption that the user can manually specify physically and semantically valid trajectories. This expectation imposes a burden on the user. Specifying complex details of a single motion, such as the kinematic arc of a hinged door or the acceleration of an object under gravity, is often counterintuitive. Furthermore, outlining dense trajectories for interactive or causal motions, such as synchronizing multiple object movements or triggering chain reactions, is also challenging. Consequently, the guidance signals provided by users are inevitably sparse, raw, and approximate, as shown by the human-annotated benchmark (Zhang et al., 2025c), whereas trajectories from point-tracking datasets (Chu et al., 2025) contain significantly more details of natural movement.

Crucially, existing models fail to account for this gap. Current motion control mechanisms (Chu et al., 2025; Geng et al., 2025; Li et al., 2025b; Zhang et al., 2025b) operate as literal executioners that move pixels as directed by users, rather than simulating realistic motion. They strictly process these sparse, raw trajectories as ground truth. This approach is suboptimal because it ignores the physical and logical gaps left by imperfect user inputs. Rather than strictly adhering to user input, it is often better to treat it as a rough

intention. A robust system must go beyond the drawn path to plan and reason about implicit consequences, anticipating the ripple effects, collisions, and secondary dynamics that the user implies but does not explicitly specify.

This limitation highlights a critical gap in the current literature: the lack of reasoning about visual context in motion control. As a result, generated videos sometimes violate physical reality or fail to capture causal effects. As the examples shown in Figure 1, consider the prompt "lift the hand blocking the dominoes"; while the user provides a trajectory to lift the hand, they implicitly expect the dominoes to fall in a chain reaction once the constraint is removed. Similarly, "turning the minute hand of a clock" implies a coupled mechanical motion where the hour hand moves in synchronization. Though proponents position video diffusion models as world simulators with emergent physical understanding (Wiedemer et al., 2025), our experiments demonstrate that motion-controlled generation can struggle with implicit causal reasoning. While models faithfully execute specified trajectories, they fail to trigger consequential dynamics. The hand lifts, but the dominoes remain frozen.

In this paper, we propose **MotiMotion**, a novel framework that integrates the reasoning capabilities of Visual Language Models (VLMs) into a motion-controlled video generation model. By leveraging world knowledge learned from large-scale pretraining, VLMs understand the provided visual contexts and reason about the underlying logic that *motivates* the user's specified motion. We formulate the framework as a reasoning-then-generation pipeline. Our key insight is that complex video dynamics can be decomposed into a training-free planning stage and a generation stage. During the planning stage, VLMs translate user input into practical control commands, including detailed narrations and physically plausible trajectories. These controls then guide the diffusion process during the generation stage, synthesizing videos aligned with world knowledge, common sense, and physical principles as shown in Figure 1.

Specifically, MotiMotion comprises two core components: a context and motion reasoner based on Visual Language Model (VLM) and a confidence-aware motion-controlled image-to-video generator. The reasoner leverages VLM's semantic understanding to refine sparse user inputs and plan dense, causally consistent motion trajectories. To further enhance the naturalness of generated motion, we introduce a confidence-aware control. We assign a score to each trajectory that indicates users' confidence in it. The score instructs the model to strictly adhere to high-confidence plans while allowing it to rely on its internal generative priors in lower-confidence regions. By grounding motion generation in explicit reasoning and adaptive control, our model relieves the user of the burden of manual simulation while retaining precise controllability.

To evaluate MotiMotion, we curate a motion-controlled image-to-video benchmark, MotiBench, composed of images with manually annotated trajectories and prompts. The samples are selected specifically to target scenarios that may need world knowledge and reasoning. Automatic VLM-based evaluation and human studies on MotiBench demonstrate that MotiMotion outperforms existing motion control methods in physical realism and logical plausibility. Ablation studies confirm the critical contributions of VLM-based prompt reasoning and motion refinement.

Our contributions are summarized as follows:

- We identify the challenge of modeling complex motion for motion-controlled video generators and integrate VLMs to leverage their reasoning capabilities.
- We propose MotiMotion, a framework that employs VLMs to provide plausible semantic and motion control, with confidence modulation that further enhances the naturalness of motion.
- We curate a specialized benchmark for physical reasoning and causal propagation, in which MotiMotion demonstrates higher user preference than existing methods.

## 2. Related Work

**Motion-Controlled Video Generation.** While recent video generation models (DeepMind, 2025c; OpenAI, 2024; Wan et al., 2025) have demonstrated unprecedented visual quality and fidelity to textual prompts, motion conditions enable precise spatial and temporal control of visual dynamics. Existing work enforce adherence to trajectory paths collected from sparse input (Ardino et al., 2021; Chen et al., 2023; Hao et al., 2018; Namekata et al., 2025; Niu et al., 2024; Wang et al., 2023; 2024c; Yin et al., 2023; Zhang et al., 2025b), point tracking (Chu et al., 2025; Feng et al., 2025; Geng et al., 2025; Gu et al., 2025; Li et al., 2025b; Zheng et al., 2025; Zhou et al., 2025) or optical flows (Burgert et al., 2025; Shi et al., 2024; Xiao et al., 2025; Xing et al., 2025; Zhang et al., 2025b;c). Instead of points or pixels, other approaches track entity movement according to semantic boundaries (Jain et al., 2024; Li et al., 2025a; Ma et al., 2024; Wang et al., 2025b; 2024a; Wu et al., 2024a;c), human pose (Hu et al., 2024; Xu et al., 2024) or camera pose (He et al., 2025; Watson et al., 2025). Motion control can also be applied to video editing (Burgert et al., 2026; Lee et al., 2026b) and real-time interaction (Shin et al., 2026). Despite the diversity across modalities and applications, these approaches assume that input trajectories perfectly capture real-world motion dynamics. Consequently, models that rigidly adhere to user inputs can produce unnatural artifacts when provided guidance is sparse, inaccurate, or physically inconsistent. In contrast, our approach treats user input as high-level intent rather than a rigid command, enabling the generation framework to adaptively adjust the strictness of

trajectory following and to synthesize realistic motion.

**Generation with Reasoning and Planning.** The integration of reasoning into generative models can be categorized into unified and modular paradigms across both image and video domains. In image synthesis, unified architectures (Chen et al., 2025c; Deng et al., 2025; Tong et al., 2025; Wu et al., 2025a;b; Xie et al., 2025) perform visual understanding and generation within single frameworks, enabling capabilities such as reasoning before generation (Deng et al., 2025; Fang et al., 2025) and subject consistency (DeepMind, 2025b; OpenAI, 2025). Conversely, modular image frameworks (Chen et al., 2025b; Guo et al., 2025; Liu et al., 2025; Wang et al., 2025c; 2024b; Wu et al., 2024b; Yao et al., 2024) decouple the cognitive load, utilizing Large Language Models (LLMs) to decompose complex prompts into explicit layouts (Feng et al., 2023) or chain-of-thought plans (Yang et al., 2024) that guide diffusion-based renderers.

This categorization extends to video generation. Unified models (Fei et al., 2024; Wei et al., 2026; Wu et al., 2025c) treat video synthesis as next-token prediction, achieving emergent reasoning through massive scale. Modular video frameworks employ LLMs to plan semantics (Huang et al., 2025; Kizil et al., 2026; Lin et al., 2024; Tian et al., 2024; Yang et al., 2025), perform iterative refinement (Lee et al., 2026a; Soni et al., 2024; Xue et al., 2025), or ensure long-horizon temporal consistency (Huang et al., 2026). Our work aligns with the modular video paradigm, which offers distinct advantages in explainability and editability over opaque unified models. By decoupling the reasoning module, we allow users to inspect, understand, and intervene in the intermediate motion plans before pixel generation, a critical feature for professional workflows.

**Physics-Aware Generation.** One line of work enforces physical realism by conditioning video generation on explicit physical cues or user interactions, such as integrating physics solvers (Li et al., 2025c; Wang et al., 2025a; Yu et al., 2025), training on synthetic data (Gillman et al., 2025; 2026), or distilling object-level physical properties from video models (Zhang et al., 2024). While our approach shares the goal of generating realistic secondary motion in response to user input, we model continuous motion that goes beyond single forces and tackle broader applications beyond physics problems, including tool mechanisms and common sense.

Another stream aligns generative models with physical laws through feedback-based optimization, such as through Direct Preference Optimization (Ji et al., 2025; Rafailov et al., 2023), geometric feedback (Yan et al., 2025), representation alignment with foundation models (Zhang et al., 2025a), or inference-time alignment (Yuan et al., 2026). While these approaches improve physical plausibility, they implicitly bake physics into the training or inference process and lack

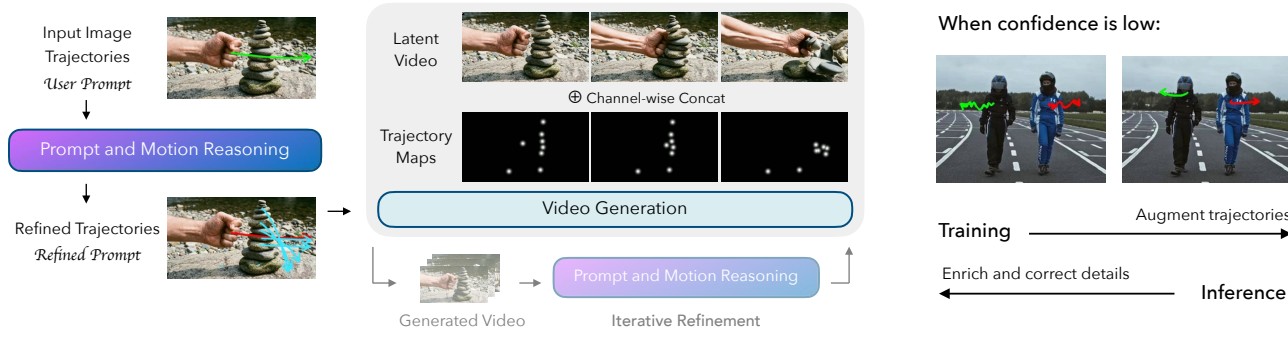

**(a) MotiMotion**  **(b) Confidence-Aware Control**

*Figure 2.* **MotiMotion Pipeline.** The core of MotiMotion is a reasoning-then-generation framework that transforms raw, sparse inputs into rich, detailed control that captures realistic world dynamics. (a) Given an input image, trajectory visualization, and textual prompt, a Visual Language Model (VLM) reveals user intention and models motion dynamics aligned with context by refining prompts and proposing trajectories for subsequent events. The reasoning process can be extended to run iteratively until satisfying users or the VLM. (b) We further enhance motion realism by introducing confidence into motion control, enabling the video generator to balance trajectory following with its own generative prior. With low confidence, the video generator considers trajectories as rough guidance and provides more details to mimic natural movements.

controllability or editability. In contrast, our modular approach integrates VLMs as robust physics reasoners and offers flexible spatial-temporal control.

## 3. Method

Our goal is to develop an intelligent motion-controlled video generation framework that bridges the gap between abstract user intentions and physically plausible visual dynamics. As illustrated in Figure 2(a), the framework comprises two synergistic components: a VLM-based reasoning module that expands sparse user inputs into dense motion plans, and a flow-matching video generator that robustly follows these plans across varying levels of precision.

In the following sections, we detail the base generator in Section 3.1, the reasoning framework in Section 3.2, and the confidence mechanism in Section 3.3.

### 3.1. Motion-Controlled Video Generation

We adapt the Wan (Wan et al., 2025) framework to support precise motion control by conditioning the generation process on dense point trajectories. Inspired by an existing design (aigc apps, 2025), we represent trajectories as spatiotemporal heatmaps and inject them into the latent space of the diffusion transformer.

**Base Video Generator.** Our method builds upon Wan, a video generation model trained with a flow-matching objective (Lipman et al., 2023), which learns a vector field $v_t$ that defines a probability path $p_t$ between a noise distribution $x_1$ and the data distribution $x_0$. The model minimizes the objective:

$$\mathcal{L}_{\text{FM}} = \mathbb{E}_{t,x_0,x_1}[\|v_t(x_t) - (x_1 - x_0)\|^2] \quad (1)$$

where $x_t = (1 - t)x_0 + tx_1$ represents a linear interpolation at time $t \in [0, 1]$. The architecture utilizes a 3D variational autoencoder (VAE) (Kingma & Welling, 2013) to map videos into a latent space, a T5 text encoder (Raffel et al., 2020) for semantic conditioning, and a Diffusion Transformer (DiT) (Peebles & Xie, 2023) as the backbone. In the image-to-video (I2V) setting, the DiT receives the concatenated noisy latent and the reference image latent. These inputs are processed by 3D self-attention (Vaswani et al., 2017) and cross-attention layers, which incorporate text embeddings as conditioning.

**Motion Representation.** In contrast to previous works that represent motion using latent embeddings (Geng et al., 2025) or frame features (Chu et al., 2025), we find that binary point maps are effective for our task. Formally, given a video of resolution $H \times W$ and length $L$, we represent $N$ point trajectories $T \in \mathbb{R}^{N \times L \times 2}$ within a zero-initialized volume $M \in \mathbb{R}^{L \times H \times W \times 3}$. For each trajectory $n \in \{1, \ldots, N\}$ and each temporal index $l \in \{1, \ldots, L\}$, we place the mean of a 2D Gaussian map at the coordinate $T(n, l)$ within the corresponding frame $M(l)$. We scale the Gaussian's standard deviation $\sigma$ relative to the video resolution and normalize the peak value to 1. Finally, the resulting map is duplicated across three channels to match the input dimensions of the VAE. This process ensures that each trajectory is explicitly mapped to the spatio-temporal coordinates of the video volume.

**Motion Condition.** To integrate motion into the generator, we first project the motion volume $M$ into the latent space using the pre-trained VAE encoder, resulting in a motion latent $z_m$. We then concatenate $z_m$ with the noisy latent $z_t$ and the reference image latent along the channel dimension before passing them to the DiT. To accommodate the

increased input dimension, we expand the weights of the DiT's initial tokenization layer. This design allows the transformer to attend to motion cues alongside structural and semantic information from the very first layer.

**Optimization.** We fine-tune the video generator to learn the correspondence between the injected motion latents and the resulting video dynamics. Specifically, we update the expanded tokenization layer and all self-attention layers within the DiT blocks while keeping the remaining parameters frozen. The new weights in the tokenization layer are initialized to zero, ensuring that the model initially relies on its pre-trained knowledge before gradually adapting to the motion conditioning. The training follows the original flow-matching objective, minimizing the error between the predicted and ground-truth vector fields.

### 3.2. Generating Plausible Motion via VLM Reasoning

In practice, user-provided motion inputs are often sparse, lack temporal pacing (e.g., constant velocity), or ignore secondary physical effects such as gravity, collisions, and environmental interactions. Paradoxically, faithful execution of these imperfect trajectories produces unrealistic, low-quality videos. To achieve motion control that aligns with visual context, we propose an intelligent framework that leverages Vision-Language Models (VLMs) to augment and refine input trajectories. Our approach exploits VLMs' world knowledge and visual reasoning capabilities to bridge the gap between sparse user inputs and plausible motion.

**Prompt and Motion Reasoning.** Our framework treats raw user input as an initial motion specification that the VLM interprets and enriches. The input to the VLM consists of three primary components: (1) motion trajectories expressed by sequences of 2D coordinates (normalized to a $[0, 1]$) as text; (2) the input image overlaid with a rendering of the trajectories (with points for positions and arrows for directions); (3) an optional text prompt. By combining these modalities, the VLM can resolve visual ambiguities, such as determining directions along long paths, using numerical ground truth and the user's textual intent.

As illustrated in Figure 2(a), the VLM reasons over these three inputs and generates a detailed narrative prompt and a refined trajectory set. The detailed narrative prompt describes the cause and effect, detailing not just the primary motion but also secondary consequences such as deformations, splashes, or lighting shifts. The refined trajectory set corrects or complements motion dynamics. It consists of (1) *refined user trajectories*, which preserve the user's spatial intent while adjusting temporal spacing to reflect physical forces like friction or acceleration; and (2) *secondary trajectories*, which represent the movement of reacting objects or static anchors identified through visual reasoning. We release the full instructions on the website.

**Iterative Selection and Correction.** We empirically find that VLMs not only predict motion from static images and trajectories but also understand motion and can judge the naturalness of motion in videos. Therefore, we can also ask VLMs to watch generated videos and further refine the previous prediction. Users can choose to perform several refinement steps until obtaining satisfying results, or the VLM declares perfect plausibility and no longer proposes secondary trajectories.

### 3.3. Learning from Imprecise Motion Guidance

While the reasoning framework significantly improves the semantic and physical logicality of motion plans, it does not guarantee pixel-level precision. VLM-predicted trajectories sometimes suffer from spatial jitter or inaccuracies due to the resolution limits of vision encoders. Similarly, manual user inputs are typically sparse, discrete, and over-smoothed, as users prioritize high-level intent over the tedious specification of natural motion dynamics.

Training a video generator to strictly follow such imperfect guidance creates a fundamental conflict: the model is forced to unlearn its pretrained knowledge of natural dynamics to overfit to synthetic artifacts. Our key insight is that the pretrained video generator already possesses a strong prior. Therefore, the conditioning mechanism should be *elastic*. It adheres tightly to the input when the trajectory is reliable, but degrades gracefully to a "rough guidance" mode when the input is imprecise, allowing the generative prior to hallucinate the missing natural details.

**Confidence-Aware Motion Control.** To formalize this, we introduce a confidence-aware training strategy that simulates the imperfections of human and VLM inputs. We assign a confidence score $s \in [0, 1]$ to each training trajectory, where $s = 1$ denotes ground-truth quality and $s \to 0$ denotes increasing unreliability. As shown in Figure 2(b), we apply degradation to low-confidence trajectories, including affine transformations (to simulate spatial uncertainty), linearization (to simulate temporal sparsity), and Savitzky-Golay smoothing (to simulate over-smoothed manual input). This approach ensures that, while low-confidence samples are likely to be noisy, the model learns a robust mapping that correlates confidence scores with expected input fidelity. Detailed configuration is discussed in Appendix A.1.

**Signal Modulation.** We convey this confidence level to the generator by modulating the strength of the motion conditioning signal. The Gaussian kernels $G$ in all frames representing a trajectory with score $s$ are scaled before being placed in the motion volume $m$: $G' = s \cdot G$. A high score produces strong peaks in the latent motion feature $z_m$, forcing the model to focus heavily on the provided coordinates. A low score dampens the signal, allowing the model to deviate from the input and encouraging it to rely

on its generative prior to synthesize naturalistic motion that roughly aligns with the user's intent, but not rigidly.

# 4. Experiments

## 4.1. Implementation Details

We develop motion control for Wan 2.2 I2V-A14B (Wan et al., 2025). The model is first trained on OpenVid (Nan et al., 2025) for motion control for 5K steps, with a learning rate of $10^{-5}$ and a batch size of 16. Then we finetune the model for confidence-aware control by degrading the trajectories of $50\%$ of the samples over the next 3K steps. We use Gemini 3.1 Pro (DeepMind, 2025a) as the motion reasoning agent.

We obtain point trajectories from the training videos using CoTracker3 (Karaev et al., 2025) with a $64 \times 64$ grid. We randomly sample $[1, 500]$ trajectories in the first stage for pure motion conditions and $[1, 20]$ trajectories in the second stage to learn confidence-based control.

## 4.2. MotiBench

**Construction.** To evaluate motion-controlled image-to-video generation under physically grounded and commonsense-driven settings, we construct a dataset, **MotiBench**, of pre-event physical interaction scenes. Each image depicts a moment immediately before a physical event, in which a small, localized action is expected to trigger a larger physical response. Examples include a hand about to cut a suspended string, a magnet being moved away from metal objects, a hair dryer approaching lightweight materials, an unstable stack of objects about to collapse, or a mechanical latch about to be released. All images explicitly capture the pre-event state, in which no visible motion has yet occurred, yet the physical configuration strongly implies an imminent interaction.

**Sources and Task Formulation.** MotiBench contains 62 pre-event images. Some images are generated using text-to-image models with carefully designed prompts, while others are collected from existing online or real-world photographs that naturally exhibit pre-event configurations. For each image, we manually design a hand-drawn motion trajectory and a corresponding textual prompt that specifies only coarse user intent (e.g., cut, pull away, rotate). The subsequent dynamics are therefore highly underdetermined and must be inferred by the model through physical and commonsense reasoning. Successful generation requires understanding changes in support relations, invisible forces, and multi-object interactions, and synthesizing physically plausible dynamics such as falling, rolling, collision, airflow-driven motion, and chain reactions.

**Taxonomy.** MotiBench spans diverse trigger types, in-

*Table 1.* **Quantitative Evaluation.** We compare MotiMotion with previous motion-controlled video generation methods on MotiBench. Our method achieves the highest scores across physical realism (physical), photorealism (photo), and semantic consistency (semantic) judged by a VLM.

| Method | Physical↑ | Photo↑ | Semantic↑ |
|---|---|---|---|
| MagicMotion (Li et al., 2025a) | 0.157 | 0.550 | 0.343 |
| Wan-Move (Chu et al., 2025) | 0.218 | 0.483 | 0.511 |
| MotiMotion | **0.302** | **0.520** | **0.665** |

*Table 2.* **Two Alternative Forced Choice (2AFC) Test**. We evaluate MotiMotion on MotiBench with a 2AFC test with a VLM-based automatic evaluator and a human study. We present **preference rate** (%) of MotiMotion against two baselines (the higher the scores are, the better our method performs against the baseline). Our method is highly preferred (with scores much larger than random chance of 50%) across both Object Property (Obj.) and Interaction (Int.), as well as overall and human evaluations.

| against Baselines | Win Rate of MotiMotion (%) | | | |
|---|---|---|---|---|
| | Obj. | Int. | Overall | Human |
| MagicMotion (Li et al., 2025a) | 72.9 | 80.8 | 78.0 | 97.9 |
| Wan-Move (Chu et al., 2025) | 71.5 | 75.0 | 73.8 | 81.4 |

cluding support failure, connection removal, non-contact force disappearance, external force onset, pressure release, mechanical release, and unstable equilibria, enabling systematic evaluation of physical plausibility, causal consistency, and controllability under minimal motion specification. More examples can be found in Section B.1

## 4.3. Evaluation

**Protocol.** To rigorously assess the physical fidelity of generated videos, we employ an automated VLM judge and a human user study. Following previous work (Chen et al., 2025a; Li et al., 2025c; Wang et al., 2025a), we evaluate physical realism, photorealism, and semantic consistency using a VLM. In addition, we utilize a Two-Alternative Forced-Choice (2AFC) protocol with ordinal strength-of-preference scoring. The 2AFC test evaluates two aspects of physical fidelity:

- *Object property*: Judges assess whether the object adheres to implied mass, gravity, and stiffness (e.g., rigid vs. soft-body deformation).
- *Interaction*: Judges determine if collisions, contact forces, and functional mechanisms (e.g., triggers, fluid flows) result in physically correct environmental changes.
- *Overall*: Holistic assessment where the judge determines the superior generation based on overall plausibility.

We utilize Gemini 3.1 Pro as an automated reasoning judge. For each test sample, the model is presented with the initial context frame, the ground-truth trajectory visualization, the

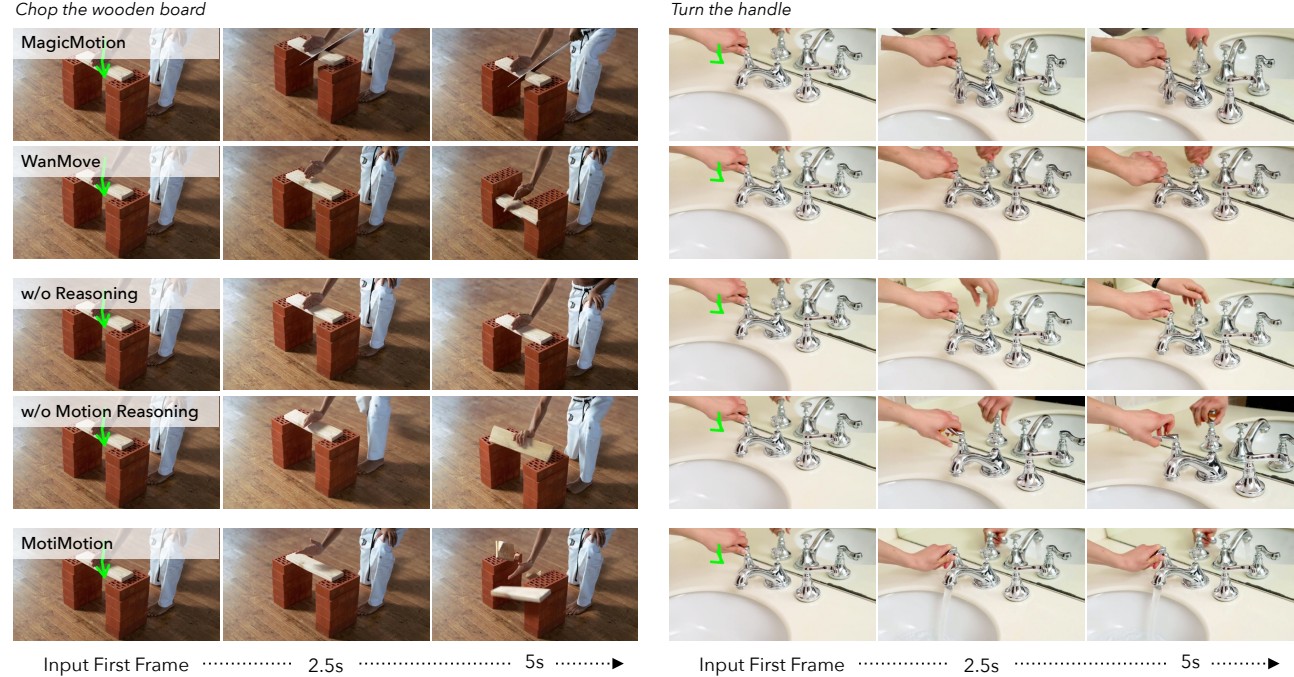

*Figure 3.* **Qualitative Comparison.** We compare MotiMotion against MagicMotion (Li et al., 2025a), Wan-Move (Chu et al., 2025), our approach without either reasoning, and without motion reasoning. MotiMotion demonstrates understanding of physics and tool mechanisms that align with the context.

*Table 3.* **Cross-Method Verification.** We apply reasoning to other motion-controlled video generation methods and find that it consistently improves their performance, especially physical realism and semantic consistency.

| Method | Physical↑ | Photo↑ | Semantic↑ |
|---|---|---|---|
| MagicMotion (Li et al., 2025a) | 0.157 | **0.550** | 0.343 |
| + Reasoning | **0.199** | 0.528 | **0.427** |
| Wan-Move (Chu et al., 2025) | 0.218 | 0.483 | 0.511 |
| + Reasoning | **0.283** | **0.488** | **0.588** |

*Table 4.* **Ablation Study of Reasoning and Control.** Each component consistently improves physical realism, photorealism, and semantic consistency.

| Method | Physical↑ | Photo↑ | Semantic↑ |
|---|---|---|---|
| Motion-Controlled Generator | 0.166 | 0.389 | 0.337 |
| + Prompt Reasoning | 0.237 | 0.475 | 0.544 |
| + Motion Reasoning | 0.285 | 0.493 | 0.641 |
| + Confidence-Aware Control | **0.302** | **0.520** | **0.665** |

*Table 5.* **Ablation Study of Reasoning without User Prompts.** Prompt reasoning takes only the input image and trajectories, yet still substantially improves performance.

| Method | Physical↑ | Photo↑ | Semantic↑ |
|---|---|---|---|
| Image + Trajectories | 0.177 | 0.353 | 0.272 |
| + Prompt Reasoning | **0.229** | **0.452** | **0.473** |

prompt, and a pair of generated videos (ours vs. baseline). The model is instructed to assign a winner and a confidence level (slight, moderate, or strong) for each metric.

**User Study.** To validate our automated metrics, we conduct a parallel user study. The protocol mirrors the automated setup, where users compare two generated videos side by side. For each pair, participants select a winner and indicate the magnitude of the difference.

**Weighted Preference Rate.** To quantify the ordinal strength labels into a scalar metric, we compute a weighted preference score. We assign weights $w \in \{1, 2, 3\}$ to slight, moderate, and strong wins, respectively (with ties = 0). The final win rate for an approach is calculated as the normalized share of the total preference. This formulation penalizes models that win only by "slight" margins while rewarding models that achieve "strong," physically distinct improve-

ments. Detailed instructions and formulation are described in Appendix Section B.3.

### 4.4. Results

**Prompt and Motion Reasoning.** We compare MotiMotion with MagicMotion (Li et al., 2025a) and Wan-Move (Chu et al., 2025) on MotiBench. The VLM evaluation in Table 1 shows that MotiMotion achieves the strongest overall performance across physical realism, photorealism, and semantic consistency. The preference results in Table 2 further

*Lift the hand*

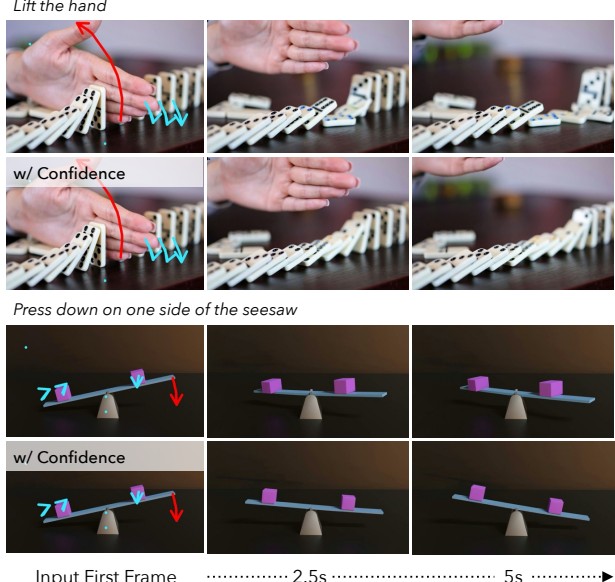

*The hand moves the minute hand around once.*

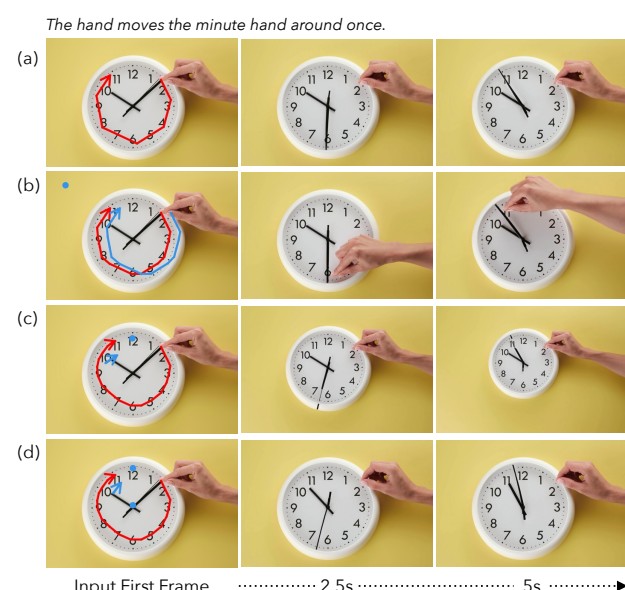

*Figure 4.* **Confidence-Aware Control** corrects artifacts caused by mismatched predicted trajectories, including dominoes falling in a deviated direction and a seesaw bent unnaturally. Red: user trajectories, Blue: VLM predicted trajectories.

*Figure 5.* **Iterative Correction.** We demonstrate the improvement made by the iterative reasoning-generation loop. Iterative corrections are applied sequentially: (a) → (b) → (c) → (d). (a) The video generator fails to model the clock dynamics with the user trajectory (red). (b) The VLM first adds a new trajectory (blue) to control human hand movement, but (c) removes it later and draws a new one that brings the hour hand to 11. After realizing the video generator misinterprets it as zoom-out, (d) it adds a fixed static point at the clock center, and the generator successfully produces a natural clock.

indicate that MotiMotion is consistently favored in both automatic and human evaluation. Qualitative comparisons in Figure 3 support this finding, showing that our method produces more physically plausible outcomes and better captures the intended interactions.

**Cross-Method Verification.** To test whether our reasoning module is broadly useful, we apply it to other motion-controlled video generation methods. As shown in Table 3, reasoning consistently improves their performance, particularly in physical realism and semantic consistency.

### 4.5. Ablation Study

**Reasoning and Control.** We analyze the effect of reasoning design and confidence-aware control. The comparison in Table 4 shows that both reasoning and confidence-aware control improve the performance. Additionally, we show two examples in Figure 4 where this control mechanism improves physical realism when the predicted trajectories do not perfectly match the visual context. In the first example, the trajectories curve downward too much, causing the dominoes to move in an unnatural direction. In the second example, the imprecise trajectories distort the seesaw. Both artifacts are corrected by confidence-aware control.

**Prompt Reasoning Design.** We further distinguish our prompt reasoning from the prompt extension or rewriting commonly used in video generation. Standard prompt extension rewrites text into a richer description, whereas our method reasons jointly over the image, trajectories, and

optional text prompt to infer user intent and potential subsequent events. This difference is important because motion-controlled generation often begins with sparse trajectories rather than requiring users to describe motion in detail. As shown in Table 5, even without user text prompts, prompt reasoning still substantially improves physical realism and semantic consistency, suggesting that the gain comes from reasoning over image- and trajectory-conditioned inputs.

**Iterative Correction.** We present an example of iterative correction by instructing the VLM multiple times to reason over generated videos, rewrite prompts, and propose necessary trajectories. As shown in Figure 5(a), pure motion control fails to model clock dynamics according to the user trajectory visualized as a red line. In (b), the VLM adds a new trajectory (blue line) to indicate human hand movement, since it believes the human hand should rotate the minute hand. In (c), the VLM correctly identifies the hour hand movement with a blue line from 10 to 11 and a fixed static point at 12, but the video generator interprets it as a zoom-out camera movement. Finally, in (d), the VLM adds another static point at the center of the clock, which signals the static camera, and the blue line successfully brings the hour hand to 11. To clarify, iterative correction is applied only in this example, not in the other experiments.

## 5. Conclusion

In this work, we devise a motion-controlled video generation framework that reasons over user intention, understands visual context, and generates plausible motion. The core of our method is a reasoning-then-generation pipeline that predicts motion and transforms raw trajectories into natural dynamics. The framework demonstrate thorough understanding of world knowledge and is highly preferred in user studies and by VLM agents. We believe this work establishes a critical foundation for achieving intelligent, realistic interaction in the digital world.

## Impact Statement

While this work advances the fidelity and control of video synthesis, we acknowledge the broader societal implications inherent to generative modeling. Enhanced motion control could potentially be misused to create deceptive or harmful content. To address these risks, we advocate for a multi-layered approach that integrates digital watermarking, rigorous content filtering, and restricts model distribution. Ultimately, fostering public digital literacy remains a critical defense against the misuse of synthetic media.

## Acknowledgments

This work was supported in part by the Institute of Information & Communications Technology Planning & Evaluation (IITP) grant funded by the Korean Government (MSIT) (No. RS-2024-00457882, National AI Research Lab Project).

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

# A. Additional Implementation Details

## A.1. Trajectory Degradation

To ensure our model effectively utilizes the input confidence score discussed in Section 3.3, we introduce a Trajectory Degradation module during training. This module synthetically corrupts ground-truth trajectories based on a sampled reliability score $s \in [0, 1]$. The goal is to enforce an inverse relationship between the reliability score and the noise level: high scores preserve the original trajectory, while low scores introduce geometric and structural distortions.

We define a degradation intensity $I$. The degradation pipeline applies three sequential operations, with the magnitude of each operation scaled linearly by $I$.

**Affine Perturbation.** We first apply random affine transformations to simulate camera instability or tracking drift. The transformations are applied relative to the trajectory's centroid:

- **Rotation:** A rotation angle $\theta$ is sampled uniformly from $[-\theta_{\max} \cdot I, \theta_{\max} \cdot I]$.
- **Scaling:** Independent scaling factors for the $x$ and $y$ axes are sampled from $1 \pm \delta_{\text{scale}} \cdot I$.
- **Translation:** A global translation vector is sampled from $[-\delta_{\text{trans}} \cdot I, \delta_{\text{trans}} \cdot I]$.

**Linearization (Temporal Subsampling).** To simulate low-fidelity tracking or temporal sparsity, we reduce the trajectory to a subset of keypoints and reconstruct it via linear interpolation. The number of keypoints $K$ is interpolated based on intensity:

$$K = \lfloor L - (L - L_{\min}) \cdot I \rfloor \tag{2}$$

where $L$ is the original sequence length and $L_{\min}$ is the lower bound (set to 10% of $L$). As $I$ increases, the trajectory becomes increasingly piecewise linear, losing high-frequency motion nuances.

**Smoothing.** Finally, to mimic the effects of over-processed or dampened tracking data, we apply a Savitzky-Golay filter. The filter's window length $W$ increases with intensity:

$$W = \lfloor W_{\min} + (W_{\max} - W_{\min}) \cdot I \rfloor \tag{3}$$

where $W_{\min} = 3$ and $W_{\max}$ is set to half the trajectory length. This results in aggressive smoothing for low-confidence inputs, effectively removing fine-grained motion details.

In our experiments, we find that the model performs better at distinguishing between user and predicted trajectories in a binary setting, where we set $s = 1$ for user trajectories and $s = 0.5$ for predicted trajectories. During training, we sample $I$ uniformly from $[0.1, 1]$ if $s = 0.5$ and set $I = 0$ if $s = 1$. We set $\theta_{\max} = 5°$, $\delta_{\text{scale}} = 0.2$, and $\delta_{\text{trans}} = 30$ pixels. For the Savitzky-Golay filter, we use a polynomial order of 2.

*Table 6.* Scenario Distribution of MotiBench.

| Category | Num of Samples | Percentage (%) |
|---|---|---|
| Collision | 9 | 15 |
| Constraint Change | 17 | 27 |
| Tool Mechanisms | 8 | 13 |
| Flow | 9 | 14 |
| Common Objects | 19 | 31 |

## A.2. Training and Inference

**Optimization.** We train the video generation model with the Adam optimizer, 100 warm-up steps, a constant learning-rate schedule, and a weight decay of 0.03.

**Inference Cost.** Prompt and motion reasoning takes about 50 seconds (10K tokens, 0.07 USD). Prompt-only reasoning takes about 15 seconds (3K tokens, 0.02 USD).

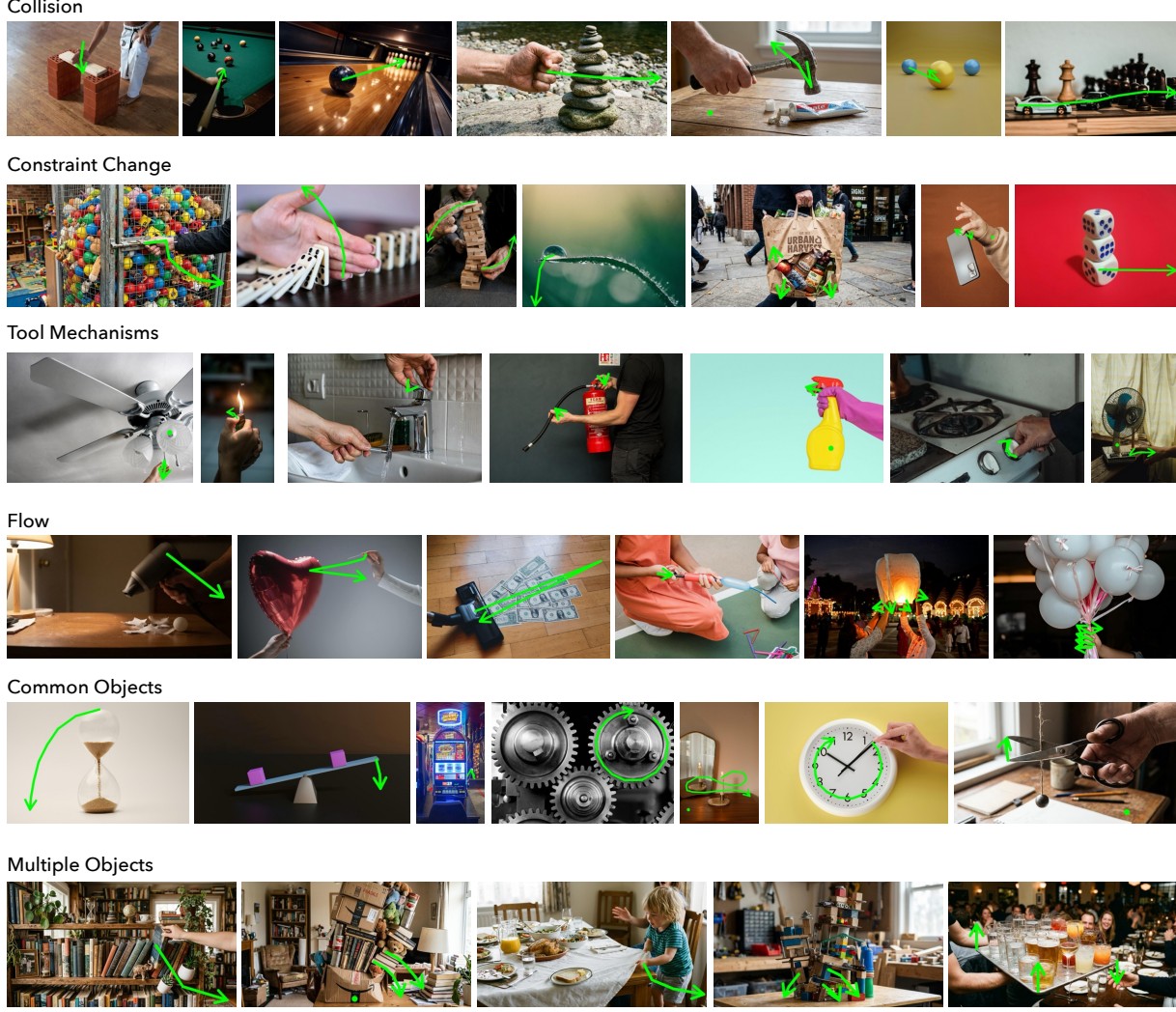

*Figure 6.* More Examples from MotiBench.

## B. Additional Experimental Details

### B.1. MotiBench

In Figure 6, we present additional examples from MotiBench across diverse scenarios, including collision, constraint change, flow, tool mechanism, and common objects, and we report the category distribution in Table 6. Among all the samples, 30% (19 samples) involve multi-object interaction.

### B.2. Additional Results

**VLM Predictions.** We show examples of trajectories predicted by the VLM. As shown in Figure 7, user trajectories are visualized in green lines, which are refined by the VLM and drawn in red. The VLM proposes new trajectories in blue.

The raw output trajectories of the first sample (the seesaw) are partially included below. The VLM is instructed to predict complete trajectories at 4 FPS from the starting to the ending frames, enabling temporal and causal ordering of multiple trajectories. In the seesaw example, the VLM proposes three new trajectories. One goes up (with $y$ decreasing), indicating the direction of the other side. Another one is a fixed static point roughly at the image center, representing the fulcrum.

User Trajectories

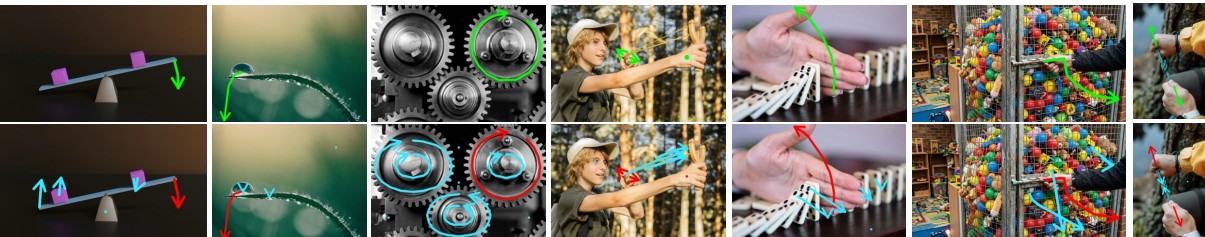

Model Trajectories

User Trajectories

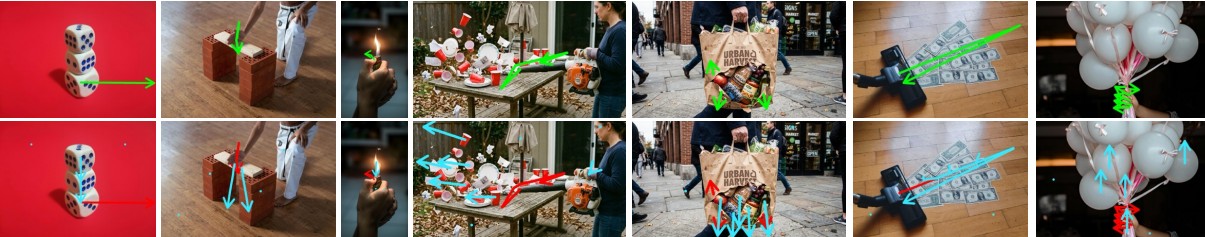

Model Trajectories

*Figure 7.* Trajectories Refined and Proposed by the VLM. Green: user input; red: refined user input; blue: model prediction.

*Table 7.* Quantitative Evaluation of Motion-Controlled Video Generation.

*(a)* 4 trajectories

| Method | FVD↓ | EPE↓ |
|---|---|---|
| Image Conductor (Li et al., 2025b) | 1735.0 | 18.92 |
| DragAnything (Wu et al., 2024c) | 1497.2 | 8.95 |
| Motion Prompting (Geng et al., 2025) | 1207.7 | 12.99 |
| Our Motion-Controlled Generator | 724.4 | 8.28 |

*(b)* 16 trajectories

| Method | FVD↓ | EPE↓ |
|---|---|---|
| Image Conductor (Li et al., 2025b) | 1838.9 | 24.26 |
| DragAnything (Wu et al., 2024c) | 1282.8 | 9.80 |
| Motion Prompting (Geng et al., 2025) | 1322.0 | 8.32 |
| Our Motion-Controlled Generator | 712.8 | 7.89 |

```
{
    "refined_user_trajectories": [
        [[0.842, 0.475], [0.846, 0.488], ..., [0.862, 0.698], [0.862, 0.711]]
    ],
    "proposed_new_trajectories": [
        [[0.190, 0.710], [0.191, 0.698], ..., [0.199, 0.492], [0.200, 0.480]],
        [[0.510, 0.750], [0.510, 0.750], ..., [0.510, 0.750], [0.510, 0.750]],
        ...
    ]
}
```

**Motion Following.** To evaluate the motion following capability of our motion-controlled video generator, we follow the protocol of Motion Prompting (Geng et al., 2025) and calculate FVD (Unterthiner et al., 2019) and end-point error (EPE) for 4 and 16 trajectories. EPE estimates the L2 distance between the conditioning trajectories and trajectories estimated from the generated videos. As shown in Table 7, our method performs comparably to previous work.

**Analysis of Failure Cases.** We observe three common failure patterns in reasoning:

- Incorrect motion prediction: inferred trajectories go in the wrong direction or follow implausible paths.
- Inaccurate grounding: the overall motion is reasonable, but trajectories are spatially shifted or distorted relative to objects and context in an image.
- Incomplete trajectories: inferred trajectories are partially correct but do not fully cover the motions needed for realistic dynamics.

We visually inspect the test set and summarize incorrect and inaccurate cases in Table 8.

*Have the female dental professional turn away from the male patient and bend down to adjust the dental chair.*

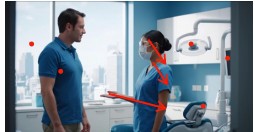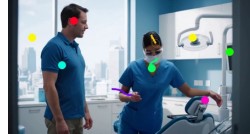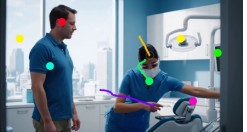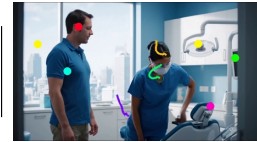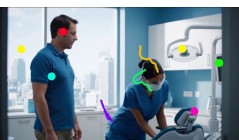

*Make the boy actively drink milk from the glass, with his mouth on the rim and eyes looking down.*

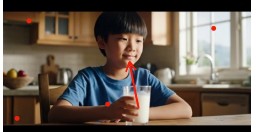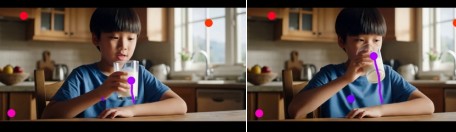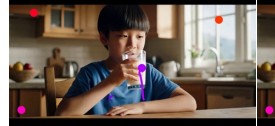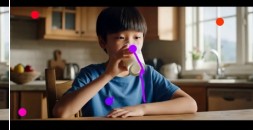

Proposed Trajectories    (a) Control with High Confidence    (b) Control with Low Confidence

*Figure 8.* Examples from MotionEdit in which the VLM proposes all trajectories, modulated with confidence-aware control.

*Table 8.* Failure Patterns.

| Failure Pattern | Percentage (%) |
|---|---|
| Incorrect | 9.7 |
| Inaccurate | 4.8 |
| Inaccurate w/ Confidence | 3.2 |

*Table 9.* Quantitative Comparison between One and Two-stage Reasoning

| Method | Physical↑ | Photo↑ | Semantic↑ |
|---|---|---|---|
| Prompt Reasoning | 0.237 | 0.475 | 0.544 |
| Prompt & Motion Reasoning (Two Stages) | **0.319** | 0.503 | 0.654 |
| Prompt & Motion Reasoning (One Stage) | 0.302 | **0.520** | **0.665** |

In Section 3.3, we train the generator with degraded trajectories to simulate imperfect inputs. The confidence value indicates when the generator should strictly follow the trajectories and when to rely more on its pretrained motion prior. Rather than treating VLM trajectories as ground truth, the generator learns to use them elastically.

This is reflected in Table 8 (Inaccurate w/ Confidence). The rate drops, indicating that MotiMotion can reduce error propagation from imperfect reasoning outputs.

For incomplete trajectories, Figure 5 shows the benefit of an optional iterative correction loop. The VLM inspects the generated video and proposes additional trajectories or static anchors in the next round.

**Reasoning without Input Trajectories.** We demonstrate the capability of motion reasoning with confidence-aware control by requiring the VLM to propose all trajectories. We use examples from MotionEdit (Wan et al., 2026), an image-editing benchmark that focuses on motion-triggered editing and motion modification in images. We prompt a VLM to propose trajectories that can guide motion in the images according to the editing instruction. Figure 8 depicts the point-tracking results for the first point of the input trajectories with (a) high and (b) low confidence. While both videos generally follow input trajectories, low-confidence control decorates motion with natural details.

**Additional Qualitative Results.** We present additional examples for comparing MotiMotion with MagicMotion, Wan-Move, our approach without either reasoning, and without motion reasoning in Figure 11.

**Analysis of Reasoning Stages.** The initial version of this paper introduces a two-stage reasoning process, as illustrated in Figure 9: (1) The VLM first refines prompts. (2) Videos are generated based on the refined prompts. (3) The VLM analyzes the videos and proposes trajectories. This process allows the VLM to complement missing motion in generated videos rather than predicting motion from images. However, compared to the one-stage reasoning we described in Section 3, this two-stage process introduces an additional reasoning and generation pass, significantly increasing latency.

After evaluating these two methods, we find that one-stage reasoning is comparable in performance while requiring much less computation and time, as shown in Table 9. We therefore opt for one-stage reasoning and conduct all experiments accordingly in this revision.

### B.3. Evaluation Details

**User Study.** We recruit 12 participants for the 2AFC user study. Each participant is tested with 30 questions sampled from all paired combinations of MotiMotion against two baselines. The screenshots of the user study interface are shown in Figure 10.

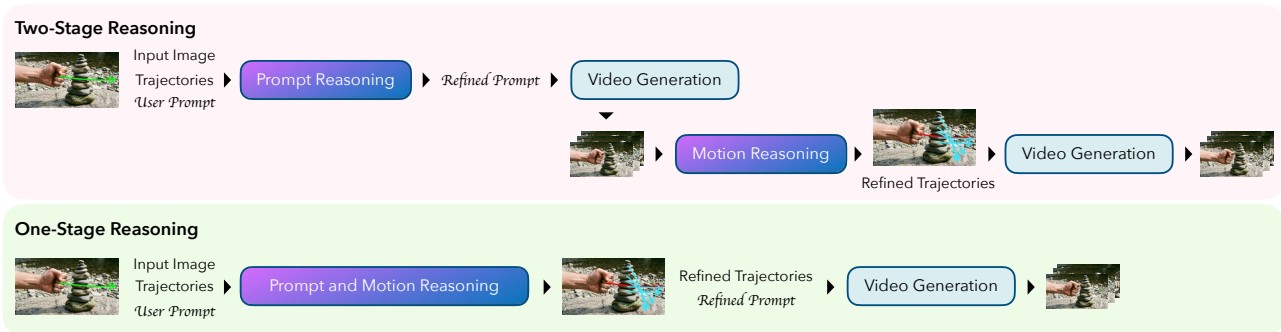

*Figure 9.* Two-Stage and One-Stage Reasoning

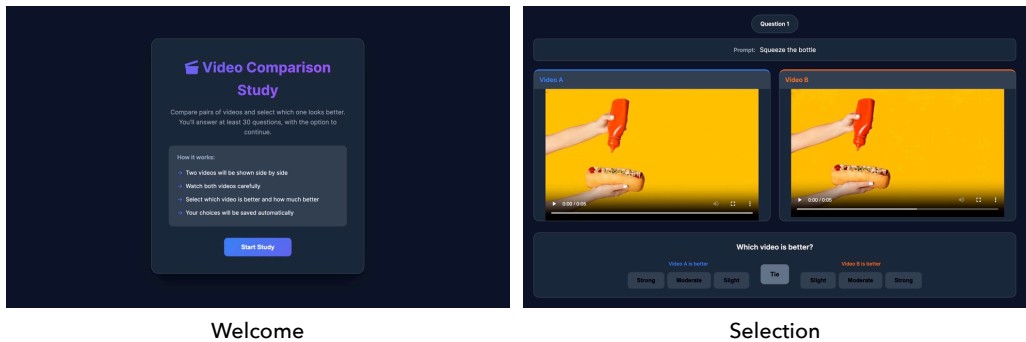

*Figure 10.* User Study Interface.

To account for the varying degrees of perceptibility in physical improvements, we adopt a weighted scoring mechanism rather than a simple binary win rate. This ensures that strong improvements (e.g., fixing a major collision failure) contribute more to the final metric than slight improvements (e.g., minor texture sharpening).

Let $\mathcal{D}$ be the dataset of pairwise comparisons. For each pair $k \in \mathcal{D}$, the evaluator (Human or Gemini) assigns a verdict $v_k \in \{\text{Tie}, \text{Win}_A, \text{Win}_B\}$ and a strength label $s_k \in \{\text{Slight}, \text{Moderate}, \text{Strong}\}$.

We define a mapping function $W(s_k)$ to assign scalar weights to the strength labels:

$$W(s_k) = \begin{cases} 1 & \text{if } s_k = \text{Slight} \\ 2 & \text{if } s_k = \text{Moderate} \\ 3 & \text{if } s_k = \text{Strong} \\ 0 & \text{if } v_k = \text{Tie} \end{cases} \tag{4}$$

To calculate the weighted preference rate, we first compute the total preference volume $S_M$ accumulated by each model $M$ (where $M \in \{A, B\}$) across all samples where it was declared the winner:

$$S_M = \sum_{k \in \mathcal{W}_M} W(s_k), \tag{5}$$

where $\mathcal{W}_M$ is the set of comparisons where model $M$ won. The final weighted win rate reported in Table 2 is calculated as the model's share of the total decisive preference volume:

$$\text{Win Rate}(A) = \frac{S_A}{S_A + S_B} \times 100\% \tag{6}$$

This metric provides a clearer signal of physical fidelity than raw win counts. For instance, a score of 65% indicates that Model A captured 65% of the total available "preference points" (where decisive "Strong" wins contribute significantly more), implying a distinct qualitative superiority rather than just a frequent but marginal advantage.

# C. Discussion

**Scale and Validity of MotiBench.** MotiBench is not intended as a general motion following benchmark or a replacement for DAVIS. It targets a complementary gap: pre-event, underdetermined interaction scenarios in which the user provides only a sparse trigger and the model must infer plausible secondary effects. This setting is not captured by motion-following benchmarks such as DAVIS, which focus on trajectory fidelity but do not evaluate whether the model can infer downstream consequences such as domino chains, gear coupling, or airflow-driven motion.

MotiBench contains 62 manually curated pre-event interaction images, comparable to those in several recent physics-aware generation benchmarks. We curate it manually because each sample must simultaneously show an imminent event and provide only coarse user intent, leaving the consequence to be inferred. This makes the benchmark compact, but targeted. It covers collision, support, and constraint changes, flow, tool mechanisms, and other multi-object interactions. The goal is not to exhaust all long-horizon settings, but to provide a focused evaluation of sparse-trigger reasoning.

The validity is supported by the evaluation and human study in Section 4.4, and cross-backbone comparison in Table 3.

**Limitation.** Latency introduced by reasoning and the dependence on external APIs may limit practical applicability, especially for real-time control. We view this as an important limitation of the current stage of the framework. Promising directions include using smaller local VLMs or distilling reasoning outputs into the video generator. More broadly, we view this work as a first step toward physically grounded and intelligent motion-following video generation by explicitly exposing the gap between trajectory following and causal motion reasoning, and by offering a practical framework to bridge it.

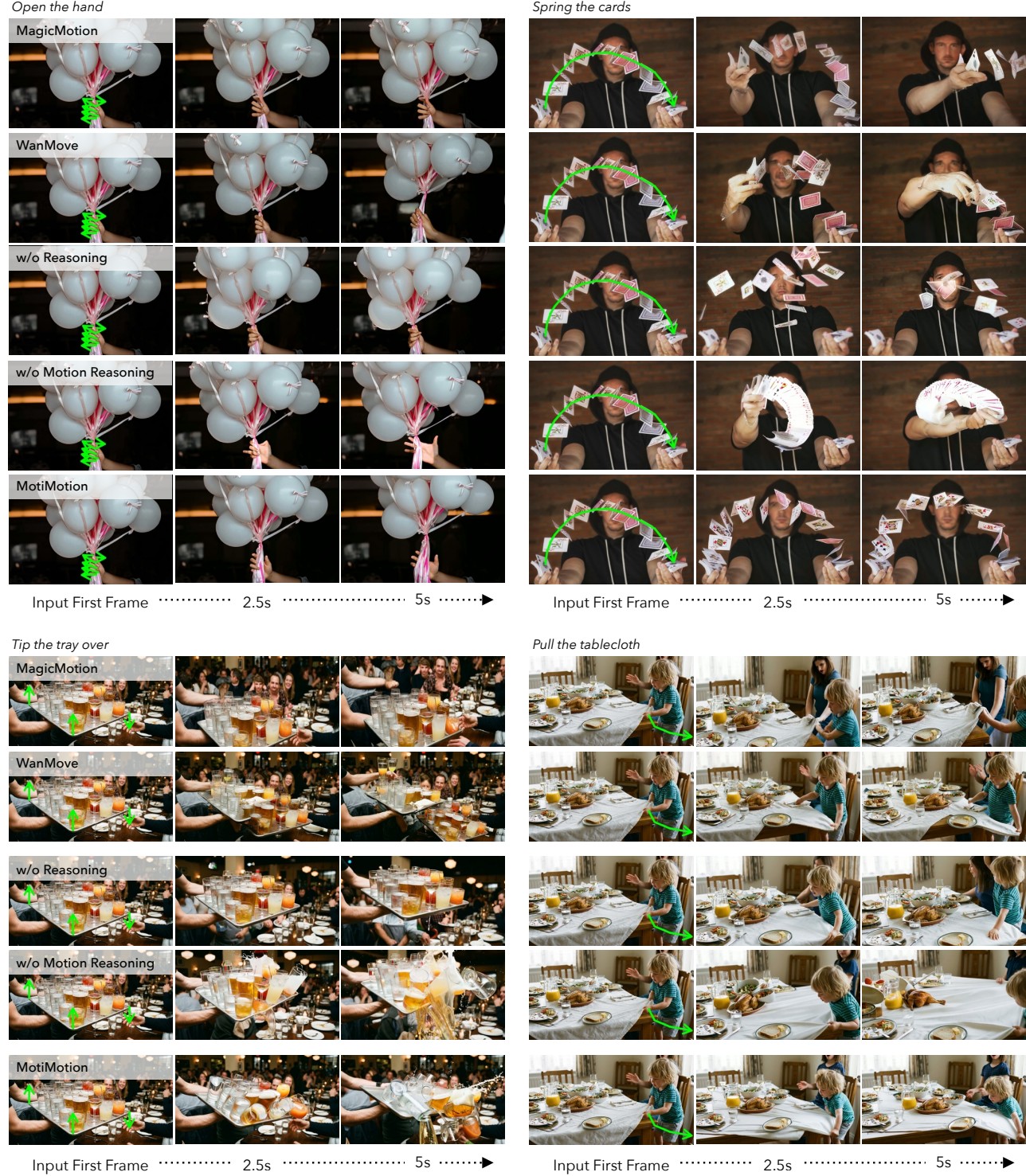

*Figure 11.* Additional Qualitative Comparison.

