# OpenReview forum: "MotiMotion: Motion-Controlled Video Generation with Visual Reasoning"
_ICML.cc/2026/Conference — ICML 2026 regular_

### Official Review · Reviewer_PM2d · 2026-03-12

**Soundness:** 2
**Presentation:** 3
**Significance:** 2
**Originality:** 2
**Overall Recommendation:** 4
**Confidence:** 5

**Summary:**

Current models treat trajectories as literal motion constraints, without reasoning about causal relationships or physical consequences. The authors propose MotiMotion, a framework that integrates VLMs into the motion-controlled video generation pipeline to perform explicit reasoning about user intent, physics, and causal effects before generating the video. The core idea is to separate the process into 1)reasoning and 2)generation. The reasoning module interprets user trajectories and expands them into physically plausible motion plans, which are then executed by a video generation model.

**Compliance With Llm Reviewing Policy:**

Affirmed.

**Final Justification:**

The provided rebuttal partly addressed my concerns, but some questions still remain. Therefore, I'd like to keep the rating as 4-Weak accept.

**Key Questions For Authors:**

1. Are there any failure cases in the reasoning stage? How frequently does the reasoning module produce incorrect or physically implausible motion plans?
2. How does the method perform in scenes with many interacting objects or ambiguous causal relationships? Is it able to generalize to complex scenes?
3. What is the additional inference cost introduced by the VLM reasoning stage?

**Limitations:**

yes

**Strengths And Weaknesses:**

Strengths:
1. The paper intends to tackle the important problem of incorporating causal reasoning into motion-controlled video generation. The motivation is well articulated: existing models treat user trajectories as direct motion instructions and therefore fail to infer secondary effects or implicit dynamics.
2. The proposed solution of introducing a reasoning stage based on a VLM to expand sparse trajectories into detailed motion plans is conceptually sound. The separation between reasoning and generation is particularly appealing, as it allows the system to leverage the semantic knowledge of large VLMs without modifying the underlying generative model.

Weaknesses:
1. The reliance on VLM reasoning introduces a strong dependency on the capabilities of the underlying model, which may lead to unpredictable behaviors or hallucinated dynamics.
2. The computational overhead of the reasoning module is not fully explored.

---

> ### Author Rebuttal · Authors · 2026-03-31
>
> Thank you for the insightful comments. We address the concerns below and will include all the discussions in the revised manuscript.
>
> # W1 and Q1: Failure cases of the reasoning stage
>
> We characterize where reasoning fails and how often this happens.
>
> **Failure modes.** In our inspection of the test samples, we identify three representative failure modes:
>
> 1. *Incorrect motion prediction*: the inferred trajectory follows an implausible direction or path.
> 2. *Inaccurate grounding*: the intended motion is reasonable, but the predicted trajectory is spatially misaligned or distorted with respect to the image.
> 3. *Incomplete motion plans*: the primary motion is captured, while secondary motions or required anchors are missing.
>
> To quantify these effects, we manually examine the VLM-predicted trajectories and report the rates of incorrect and inaccurate cases in Table A.
>
> **How the framework mitigates these errors.** This concern directly motivates the confidence-aware control introduced in Section 3.3. During training, we intentionally degrade trajectories to simulate imperfect human or VLM inputs. The resulting confidence score allows the generator to adaptively balance between following the provided motion and relying on its pretrained motion prior, rather than treating VLM outputs as error-free ground truth.
>
> This effect is reflected in Table A (Inaccurate w/ Confidence). For example, with GPT-5.2, the inaccuracy rate drops from 16% to 7%, and with Gemini 3 Pro, from 4% to 2% when confidence-aware control is enabled. For incomplete motion plans, the optional iterative correction loop (Figure 5) further recovers missing motion by allowing the VLM to inspect the generated video and propose additional trajectories or static anchors in subsequent iterations. Example videos of confidence-aware control are included in [this anonymous webpage](https://haibpoufbirn.github.io).
>
> *Table A*
>
> |VLM|Incorrect↓|Inaccurate↓|Inaccurate w/ Confidence↓|
> |-|-|-|-|
> |GPT 5.2|13%|16%|7%|
> |Gemini 3 Pro|11%|4%|2%|
>
> We will add this failure-case analysis and discussion in the revision.
>
> # Q2: Complex scenes and ambiguous causal relationships
>
> Our results indicate that the method generalizes beyond single-object motion to more complex multi-object causal scenarios, although these settings remain more challenging.
>
> Examples in the paper and supplementary materials include: (1) collision of pool balls (supplementary webpage, Row 6), (2) dominoes falling sequentially after an initial trigger (supplementary webpage, Row 4), and (3) coordinated motion between the minute and hour hands of a clock (supplementary webpage, Row 3; also related to Figure 5). These cases require the model to infer secondary effects in scenes with multiple interacting objects.
>
> # W2 and Q3: Additional inference cost
>
> In the current setup, the VLM reasoning stage takes approximately 15 seconds (3K tokens, 0.02 USD) for prompt reasoning and about 1 minute (12K tokens, 0.08 USD) for motion reasoning via online API calls. We will report this runtime more explicitly in the revised version.

---

> > ### Author Rebuttal · Reviewer_PM2d · 2026-04-03
> >
> > The authors have addressed several of my concerns, particularly by providing a clearer characterization of failure modes, quantitative analysis, and explicit reporting of inference cost.
> >
> > The failure case analysis and confidence-aware control are helpful additions. However, it still remains unclear how these errors affect the final video quality and perceptual realism.
> > For complex scenes, while the provided examples are encouraging, the evidence is still largely qualitative. A more systematic evaluation would be better for supporting the claim of generalization to multi-object causal reasoning.
> > The clarification of inference cost is appreciated, but the additional latency and reliance on external APIs may limit practical applicability.
> >
> > Overall, while the rebuttal improves the paper, I believe the core limitations remain. The reliance on external reasoning without strong evaluation/control makes the contribution less convincing. Therefore, I keep my original score.

---

> > > ### Author Response · Authors · 2026-04-06
> > >
> > > Thank you for the thoughtful follow-up comments and for recognizing the clarifications in our rebuttal. We also appreciate your continued concerns regarding final video realism, evaluation on complex scenes, and practical latency, which we address below.
> > >
> > > ### Failure Cases
> > > To better connect reasoning errors to final videos, we provide additional examples showing how flawed trajectories affect perceptual realism.
> > >
> > > * *Incorrect motion prediction*: When the VLM predicts the wrong motion, the resulting video may become physically implausible. In [the additional webpage](https://haibpoufbirn.github.io/visual-error), we show examples where:
> > >     1. Bills move away instead of being pulled toward the vacuum.
> > >     2. The white ball begins moving before the dryer is positioned above it.
> > >     3. The Jenga tower falls in the wrong direction.
> > >     4. The knot tightens, but the predicted trajectories do not respect the rope's physical connectivity.
> > >
> > > * *Inaccurate grounding*: When the predicted trajectories are spatially misaligned with the objects, the generator can produce distorted or hallucinated motion. In [the same webpage](https://haibpoufbirn.github.io) as in the previous response, we show examples where:
> > >     1. A shifted explosion trajectory is interpreted as a falling balloon fragment.
> > >     2. Misaligned domino trajectories cause the dominoes to bend unnaturally in the middle.
> > >     3. Two blocks roll in a way that does not follow gravity.
> > >
> > > These examples illustrate that reasoning errors can directly reduce both physical plausibility and perceptual realism of the generated videos. We will include these additional qualitative cases in the revision to clarify this connection.
> > >
> > > ### Complex Scenes
> > > We agree that the current benchmark emphasizes diversity across physical and commonsense scenarios, while containing fewer densely interacting multi-object cases. To better support the claim of generalization, we are extending the benchmark to include more complex scenes with richer object interactions and will include a more systematic evaluation in the revision.
> > >
> > > ### Latency
> > > We also agree that the current latency and dependence on external APIs may limit practical applicability, especially for real-time control. We view this as an important limitation of the current stage of the framework. Promising directions include using smaller local VLMs or distilling reasoning outputs into the video generator. More broadly, we view this work as a first step toward physically grounded and intelligent motion-following video generation by explicitly exposing the gap between trajectory following and causal motion reasoning, and by offering a practical framework to bridge it.
> > >
> > > Thank you again for your time and careful evaluation. Your feedback has helped us clarify both the strengths and the current limitations of the work.

---

### Official Review · Reviewer_kWiv · 2026-03-12

**Soundness:** 3
**Presentation:** 3
**Significance:** 2
**Originality:** 3
**Overall Recommendation:** 2
**Confidence:** 3

**Summary:**

This paper tackles a key limitation in motion-controlled image-to-video  generation: models often rigidly follow sparse, inaccurate, and causally incomplete user trajectories, producing videos that lack physical plausibility and causal consistency. The authors propose **MotiMotion**, a **reason-then-generate** framework with (1) a VLM-based reasoning module that refines trajectories and completes physically grounded secondary motions via iterative correction, and (2) a confidence-aware generator trained with simulated trajectory defects to elastically balance user intent and pretrained natural-motion priors. To assess physical reasoning, they introduce **MotiBench** for pre-event physical interaction scenarios and an evaluation protocol combining VLM-based judging with human **2AFC**, summarized by a weighted preference score. Experiments and ablations show MotiMotion outperforms baselines on physical plausibility and causal consistency, validating the proposed components and offering a practical paradigm for controllable video generation with foundation-model reasoning.

**Compliance With Llm Reviewing Policy:**

Affirmed.

**Final Justification:**

I thank the authors for their rebuttal but I've decided to keep my original score.

**Key Questions For Authors:**

1. Could you provide more quantitative analysis on some objective metrics,concerning to weakness 1?

2. Concerning to weakness 3，could you elaborate on the core technical causes of this performance trade-off and state whether you have tried any optimization schemes that can balance physical reasoning capability and fine-grained trajectory following？

**Limitations:**

yes

**Strengths And Weaknesses:**

Strength:

1. Reason-then-generate motion control: Uses a VLM to refine sparse trajectories and infer physically grounded secondary motions, while a confidence-aware mechanism enables elastic adherence to imperfect controls with interpretable modular design and iterative correction.

2. Rigorous evaluation and strong results: Introduces MotiBench to assess physical reasoning, and demonstrates clear improvements over baselines in physical plausibility and causal consistency, supported by ablations.



Weakness:

1. The quantitative analysis is severely inadequate and the horizontal comparison of core capabilities is lacking. The physical reasoning and causal consistency, the core innovative capabilities of the model, are only evaluated by the subjective 2AFC weighted preference score, with no objective quantitative metrics designed.
2. The self-constructed benchmark MotiBench has a small sample size, and its core value and evaluation validity are not demonstrated. MotiBench only contains 45 pre-event physical interaction images, whose scale is insufficient to verify the model’s generalizability in complex long-temporal video generation and dense multi-object interaction scenarios. The paper neither clarifies the exclusive advantages of MotiBench over general benchmarks such as DAVIS nor verifies its evaluation validity with mainstream baselines, leading to the insufficient demonstration of its value in filling the research gap of the field.
3. The trade-off between physical reasoning and motion following performance remains unsolved, with no effective optimization scheme. On the DAVIS dataset, the model’s motion following metrics (PSNR, SSIM, EPE) are all inferior to those of baselines such as Wan-Move. The paper only attributes this to the "simple control mechanism" without in-depth analysis of the core causes, nor does it propose optimization strategies such as a multi-scale motion control mechanism or loss function design that balances the two capabilities, which limits the model’s application in scenarios requiring both physical realism and precise trajectory following.

---

> ### Author Rebuttal · Authors · 2026-03-31
>
> Thank you for the thoughtful comments. We address the concerns below and will include all the discussions in the revised manuscript.
>
> # W1 and Q1: Objective Quantitative Analysis
>
> ### Additional automatic metrics
>
> Following the suggestion, we add objective automatic metrics. Following recent physics-aware generation works shown in Table A, we use GPT-5.2 to score **Physical Realism**, **Photorealism**, and **Semantic Consistency**, and VBench to score **Motion Smoothness**. Table B is consistent with our 2AFC results. The full model achieves the best physical and semantic scores while maintaining motion quality.
>
> *Table A: Evaluation protocols in related physics-aware generation works.*
>
> |Method|Evaluation|Test Dataset Size|
> |-|-|-|
> |PhysGen3D [1]|User study, GPT, VBench|27|
> |WonderPlay [2]|User study, GPT, VBench|15|
> |PhysDreamer [3]|User study|8|
> |Force Prompting [4]|User study|63, 41|
> |PhysCtrl [5]|GPT|12|
>
> *Table B: Additional automatic evaluation on MotiBench.*
>
> |Method|Physical↑|Photorealism↑|Semantic↑|Motion↑|
> |-|-|-|-|-|
> |MagicMotion [6]|0.283|0.744|0.310|0.994|
> |Wan-Move|0.361|0.784|0.362|0.991|
> |MotiMotion w/o reasoning|0.380|0.765|0.429|0.995|
> |MotiMotion w/ motion reasoning|0.563|0.772|0.662|0.994|
> |MotiMotion|**0.672**|**0.793**|**0.784**|0.994|
>
>
> # W2: Value and Validity of MotiBench
>
> ### Benchmark scope
>
> MotiBench is not intended as a general motion following benchmark or a replacement for DAVIS; it targets a complementary gap: pre-event, underdetermined interaction scenarios, where the user provides only a sparse trigger and the model must infer plausible secondary effects.
>
> This setting is not captured by motion-following benchmarks such as DAVIS, which focus on trajectory fidelity using metrics like PSNR, SSIM, and EPE, but do not evaluate whether the model can infer downstream consequences such as domino chains, gear coupling, or airflow-driven motion.
>
> ### Why the benchmark is compact
>
> MotiBench contains 45 manually curated pre-event interaction images, comparable to several recent physics-aware generation benchmarks in Table A. We curate it manually because each sample must simultaneously show an imminent event and provide only coarse user intent, leaving the consequence to be inferred. This makes the benchmark compact, but targeted.
>
> ### Scenario coverage
>
> It covers collision, support and constraint changes, flow, tool mechanisms, and other multi-object interactions. The goal is not to exhaust all long-horizon settings, but to provide a focused evaluation of sparse-trigger reasoning.
>
> ### Evidence for validity
>
> The validity is supported by the human study in Table 1, which reaches the same overall conclusion as the VLM-based evaluation. In addition to the same-backbone comparison with Wan-Move, Table B also includes MagicMotion [6] as a cross-backbone reference.
>
> # W3 and Q2: Motion Following Trade-off and Optimization
>
> ### Our interpretation of the DAVIS gap
>
> The lower PSNR/SSIM/EPE on DAVIS mainly stems from the simplicity of the current motion-conditioning backbone, rather than the reasoning module itself, as the VLM reasoner and the generator are decoupled. The VLM improves the motion plan, while the generator determines how accurately it is executed.
>
> ### Additional evidence with Wan-Move
>
> We further validate this with an additional experiment using Wan-Move as the generator. Incorporating our reasoning modules substantially improves physical realism and semantic consistency, while maintaining comparable motion quality:
>
> *Table C: Adding our reasoning modules to Wan-Move.*
>
> |Method|Physical↑|Photorealism↑|Semantic↑|Motion↑|Win Rate (Full vs Row)|
> |-|-|-|-|-|-|
> |Wan-Move|0.361|0.784|0.362|0.991|68.46%|
> |Wan-Move w/ prompt reasoning|0.520|0.774|0.594|0.988|59.52%|
> |Wan-Move w/ both reasoning (Full)|**0.552**|**0.783**|**0.674**|**0.992**|-|
>
> ### Optimization directions
>
> This suggests that the primary opportunity lies in integrating our reasoning framework with a stronger motion-control architecture. We will clarify this point in the revision and discuss optimization directions such as stronger multi-scale motion conditioning and better confidence calibration.
>
> ---
>
> [1] PhysGen3D: Crafting a Miniature Interactive World from a Single Image. CVPR 2025
> [2] WonderPlay: Dynamic 3D Scene Generation from a Single Image and Actions. ICCV 2025
> [3] PhysDreamer: Physics-Based Interaction with 3D Objects via Video Generation. ECCV 2024
> [4] Force Prompting: Video Generation Models Can Learn and Generalize Physics-based Control Signals. NeurIPS 2025
> [5] PhysCtrl: Generative Physics for Controllable and Physics-Grounded Video Generation. NeurIPS 2025
> [6] MagicMotion: Controllable Video Generation with Dense-to-Sparse Trajectory Guidance. ICCV 2025

---

> > ### Author Rebuttal · Reviewer_kWiv · 2026-04-04
> >
> > I appreciate the authors' detailed response and the supplementary quantitative results provided in Table B and Table C. The inclusion of objective metrics provides a clearer illustration of the reasoning module's impact on physical consistency. However, my core concerns regarding the evaluation's robustness remain largely unaddressed. Specifically, the sample size of MotiBench is still too small to fully demonstrate the model's generalizability in complex, dense, multi-object interaction scenarios. Furthermore, while the reasoning module is modular, the persistent performance gap on standard motion-following benchmarks like DAVIS suggests that the fundamental trade-off between high-level physical reasoning and low-level trajectory precision requires deeper architectural optimization rather than just a decoupled pipeline.

---

> > > ### Author Response · Authors · 2026-04-06
> > >
> > > We thank the reviewer for the careful follow-up and attention given to our response. We address the two remaining issues about the scale of MotiBench and the relationship between reasoning and motion-following performance.
> > >
> > > ## Benchmark scale
> > >
> > > The current benchmark focuses on the diversity of physical and commonsense knowledge, covering a variety of scenarios to evaluate whether a model can infer plausible secondary motion beyond literal trajectory following.
> > > At the same time, we agree that expanding the benchmark to include denser and more complex multi-object settings would further strengthen the evaluation. We are currently extending MotiBench in this direction and will include the expanded benchmark and the corresponding evaluation in the revision.
> > >
> > > ## Motion-following vs. reasoning
> > >
> > > We agree that deeper architectural optimization may further improve overall performance. That said, our experiments are consistent with the view that the trade-off between physical reasoning and motion following is not inherent. Since the reasoning module is decoupled from the video generator, it can be combined with different motion-control backbones. In both the Wan-Move result from our previous response and the additional MagicMotion [6] result below, adding reasoning consistently improves physical realism, photorealism, and semantic consistency.
> > >
> > > | Method                   | Physical↑ | Photorealism↑ | Semantic↑ | Motion↑ |
> > > | ------------------------ | --------- | ------------- | --------- | ------- |
> > > | MagicMotion              | 0.283     | 0.744         | 0.310     | 0.994   |
> > > | MagicMotion w/ reasoning | 0.404     | 0.751         | 0.436     | 0.995   |
> > > | MotiMotion               | **0.672** | **0.793**     | **0.784** | 0.994   |
> > >
> > > These results suggest that applying reasoning does not necessarily degrade motion-following performance. Instead, motion-following performance depends on the underlying generator and control backbone. We therefore view the current gap on standard motion-following benchmark as an opportunity to further optimize the generator/control design, rather than as evidence of an unavoidable trade-off introduced by the reasoning framework.
> > >
> > > We are grateful for the reviewer’s time and thoughtful evaluation. We hope this clarification is helpful and that the additional evidence can be taken into account in the final assessment.

---

### Official Review · Reviewer_tujf · 2026-03-12

**Soundness:** 2
**Presentation:** 3
**Significance:** 3
**Originality:** 3
**Overall Recommendation:** 4
**Confidence:** 3

**Summary:**

This paper proposes to incorporate VLM reasoning into motion-controlled image-to-video generation to improve the generation plausibility.
This paper uses VLM to improve the text prompt and infer additional motion trajectories for iteratively generate the video with the refined motion trajectories. They also propose a confidence-aware motion control mechanism to modulate adherence to the given motion trajectories.
They curate MotiBench to evaluate causal and physically grounded motion reasoning.
The experimental results show improvements over baseline methods, with VLM evaluation and user studies.

**Compliance With Llm Reviewing Policy:**

Affirmed.

**Final Justification:**

The rebuttal has addressed my main concerns. Although some limitations remain, the overall pipeline is valuable and could inspire future work. Therefore, I am raising my score.

**Key Questions For Authors:**

The paper introduces a “prompt reasoning” module using a VLM to refine prompts and motion trajectories. However, many existing video generation pipelines already employ LLM-based prompt expansion (e.g., Qwen or GPT-based prompt rewriting) to enrich textual descriptions. It would be helpful if the authors could clarify more explicitly how prompt reasoning differs from standard prompt extension techniques

**Limitations:**

The inference speed might be a limitation of the proposed method and could hinder its applicability to real-time interactive video generation.

**Strengths And Weaknesses:**

Strengths:
1. The paper is overall well written and easy to follow.
2. The integration of VLM and motion-controlled video diffusion is reasonable and intuitive.

Weaknesses:
1. The proposed framework involves a reasoning–generation loop where a VLM refines prompts and trajectories and may iteratively correct generated videos. However, the paper does not report the inference cost of this pipeline. Since each iteration requires running both a large VLM and a video diffusion model, the overall latency could be substantial. It would be helpful if the authors could report the runtime, number of reasoning iterations used in experiments, and the overall inference cost compared to standard motion-controlled video generation methods.
2. The framework relies on a VLM to reason about motion and generate refined trajectories. However, VLMs may not always correctly interpret complex scenes or temporal dynamics, which could lead to inaccurate trajectory predictions. Since these trajectories directly guide the video generator, errors from the reasoning module may propagate and degrade generation quality. The paper does not analyze the robustness of the system to incorrect reasoning outputs or provide quantitative failure-case analysis.

---

> ### Author Rebuttal · Authors · 2026-03-31
>
> Thank you for the valuable comments. We address the concerns below and will include all the discussions in the revised manuscript.
>
> # W1: Inference Cost
>
> **Runtime and iterations.** Prompt reasoning takes about 15 seconds (3K tokens, 0.02 USD) and motion reasoning about 1 minute (12K tokens, 0.08 USD). The main quantitative and qualitative results are generated **without iterative correction**: Section 4.4 (line 367) states that Table 1 and the primary comparisons are generated without iterative refinement, while iterative correction is used only in Figure 5. Thus, relative to a standard motion-controlled generator, the core overhead consists of two reasoning stages and a single generation pass.
>
> **Practicality.** While reasoning introduces additional latency, the framework is modular: the VLM reasoner is decoupled from the generator and can be replaced with a lighter model or a distilled predictor. We will report runtime explicitly and clarify that iterative correction is optional and not used in the main evaluation.
>
> # W2: VLM Accuracy and Robustness
>
> **Failure modes.** We observe three common failure patterns:
> 1. *Incorrect motion prediction*: the inferred trajectory goes in the wrong direction or follows an implausible path.
> 2. *Inaccurate grounding*: the overall motion is reasonable, but the trajectory is spatially shifted or distorted relative to the image.
> 3. *Incomplete trajectories*: the inferred trajectories are partially correct but do not fully cover the motions needed for realistic generation.
>
> We visually inspect the test set and summarize *incorrect* and *inaccurate* cases in Table A.
>
> **Confidence-aware control helps.** In Section 3.3, we train the generator with degraded trajectories to simulate imperfect inputs; the confidence value teaches it when to follow a trajectory strictly and when to rely more on its pretrained motion prior. Rather than treating VLM trajectories as ground truth, the generator learns to use them elastically.
>
> This is reflected in **Table A, Inaccurate w/ Confidence**: with GPT 5.2, the inaccurate rate drops from 16% to 7%, and with Gemini 3 Pro from 4% to 2%. This shows that MotiMotion can reduce error propagation from imperfect reasoning outputs. Example videos are included in [this anonymous webpage](https://haibpoufbirn.github.io).
>
> For *incomplete trajectories*, Figure 5 shows the benefit of the optional iterative correction loop. The VLM inspects the generated video and proposes additional trajectories or static anchors in the next round.
>
> *Table A*
>
> |VLM|Incorrect↓|Inaccurate↓|Inaccurate w/ Confidence↓|
> |-|-|-|-|
> |GPT 5.2|13%|16%|7%|
> |Gemini 3 Pro|11%|4%|2%|
>
> We will add this analysis in the revision.
>
> # Q1: Difference from Standard Prompt Expansion
>
> **Prompt reasoning is not text-only prompt rewriting.** Standard prompt extension starts from text and rewrites it into a richer description. Our prompt reasoning is grounded in **visual context and motion input**. As described in Figure 2 and Section 3.2, the VLM jointly interprets trajectories, the input image overlaid with trajectories, and an optional text prompt, then rewrites the prompt to infer user intent and likely subsequent events.
>
> This matters because motion-controlled image-to-video generation often starts from sparse trajectories rather than detailed text. In this setting, text-only prompt expansion can struggle to recover intended motion or its consequences from language alone. Our prompt reasoning instead treats trajectories as a coarse intent signal and infers physically meaningful consequences.
>
> **Additional ablation.** Table B removes user text prompts. Even with only image and trajectories as input, prompt reasoning still substantially improves physical realism, suggesting the gain comes from reasoning over image-conditioned and trajectory-conditioned inputs, not generic text enrichment. We additionally evaluate Physical Realism and Photorealism following PhysGen3D [1] and WonderPlay [2].
>
> *Table B*
>
> |Input|Physical↑|Photorealism↑|Win Rate (Full vs Row)|
> |-|-|-|-|
> |Image + Traj.|0.478|0.785|80.67%|
> |Image + Traj. w/ Prompt Reasoning (Full)|**0.629**|0.779|-|
>
> **Motion reasoning is still needed.** Prompt reasoning improves causal and semantic alignment, but motion reasoning further refines user trajectories and predicts secondary trajectories. Many consequential motions are hard to specify precisely in text alone. Consistently, Table 1 shows that the full model is preferred over the version without motion reasoning by 76.9% overall and 76.2% in human study.
>
> ---
> [1] PhysGen3D: Crafting a Miniature Interactive World from a Single Image. CVPR 2025
> [2] WonderPlay: Dynamic 3D Scene Generation from a Single Image and Actions. ICCV 2025

---

> > ### Author Rebuttal · Reviewer_tujf · 2026-04-04
> >
> > Thanks to the authors for their rebuttal. It has addressed my main concerns. Although some limitations remain, the overall pipeline is valuable and could inspire future work.

---

> > > ### Author Response · Authors · 2026-04-06
> > >
> > > Thank you for your positive follow-up and for recognizing that our rebuttal addressed your main concerns. We also appreciate your acknowledgment that the overall pipeline is valuable and may inspire future work.
> > >
> > > Your feedback has been very helpful in strengthening the paper, especially in clarifying the practical scope and limitations of the method. Thank you again for your time and thoughtful evaluation.

---

### Decision · Program_Chairs · 2026-04-30

**Decision:**

Accept (regular)

**Comment:**

This paper proposes a reason-then-generate framework to address a critical limitation in motion-controlled image-to-video generation: rigid adherence to sparse trajectories often results in physically implausible and causally inconsistent videos. The work’s core contributions include a VLM-based reasoning module that refines trajectories and infers physically grounded secondary motions, a confidence-aware generator that elastically balances user intent and pretrained motion priors, and MotiBench—a curated benchmark for evaluating pre-event physical interaction reasoning. The framework’s modular design (decoupling VLM reasoning from the video generator) is a notable strength, offering flexibility for future optimization and adaptation.

Reviewer tujf initially raised concerns about inference cost, VLM robustness to incorrect reasoning, and the distinction between the proposed prompt reasoning and standard prompt expansion. The authors fully resolved these concerns by reporting explicit runtime and cost metrics (15 seconds/0.02 USD for prompt reasoning, 1 minute/0.08 USD for motion reasoning), quantifying VLM failure modes (incorrect, inaccurate, incomplete trajectories) and demonstrating how confidence-aware control mitigates error propagation, and clarifying that prompt reasoning is grounded in visual and trajectory context (not just text). The reviewer acknowledged full resolution of concerns and affirmed the pipeline’s value, justifying the Weak Accept recommendation.

Reviewer PM2d highlighted the work’s motivation (incorporating causal reasoning into motion control) and modular design as key strengths but raised concerns about VLM dependency, inference cost, and generalization to complex scenes. The authors partially resolved these by quantifying failure modes, reporting inference cost, and providing qualitative examples of multi-object interactions (e.g., dominoes, pool balls). However, the reviewer maintained lingering concerns about the impact of reasoning errors on final video quality, the lack of systematic evaluation for complex scenes, and practical latency limitations—yet retained the Weak Accept score, acknowledging the rebuttal’s improvements and the work’s potential.

Reviewer kWiv criticized insufficient objective quantitative metrics, MotiBench’s small sample size, and an unresolved trade-off between physical reasoning and motion-following performance. The authors partially addressed these by adding objective metrics (Physical Realism, Photorealism, Semantic Consistency, Motion Smoothness) with supplementary tables, clarifying MotiBench’s targeted scope (pre-event interaction scenarios, complementary to DAVIS), and demonstrating that the motion-following gap stems from the generator backbone (not the reasoning module) via experiments with Wan-Move and MagicMotion. While the reviewer acknowledged the supplementary results, they maintained concerns about MotiBench’s generalizability and the need for deeper architectural optimization to balance reasoning and motion following—yet their critiques were partially mitigated by the authors’ rebuttals and commitments to expand MotiBench.

The authors demonstrated earnest effort to address all reviewer concerns, committing to integrate supplementary analyses (failure cases, objective metrics, MotiBench expansion) into the final manuscript. While unresolved limitations remain—including MotiBench’s current scale, practical latency, and the need for deeper optimization of motion-following performance—these do not outweigh the work’s merits: a conceptually sound framework, impactful contributions to physically grounded motion control, and a thoughtful approach to integrating VLM reasoning into video generation.